# Fast and Precise Generic Model for Position-Based Trajectory Prediction of Inland Waterway Vessels

**Navreet S. Thind \*, Justus Hering and Dirk Söffker** 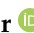

Institute of Dynamics and Control, University of Duisburg-Essen, 47057 Duisburg, Germany
\* Correspondence: navreet.singh-thind@uni-due.de

**Abstract:** Vessel motion simulation as well as model-based accurate trajectory prediction of vessels require accurate models with respect to related dynamic properties. The ability to predict vessel's trajectory behaviors will become relevant in the case of future autonomous navigation of vessels to predict the behavior of others. The definition of models or parameters can be realized via first principles or by using experimental modeling methods leading to a time invariant or variant model. Existing hydrodynamical modeling approaches are based on mathematical approaches, which use parameters like mass, hydrodynamic forces, wind velocity, depth under the keel, loading parameters, etc. So, determining a dynamic vessel's model is a complex task, since the model is vessel-specific. For collision avoidance of autonomous or assisted vessels, the trajectory prediction of encountering other vessels is especially required. It is not possible to use complex hydrodynamical models of encountering vessels online due to missing required information/measurements. Even existing deep learning approaches provide better predictions, but are still insufficient for collision avoidance in the case of strong dynamical changes, since the considered input sequences are long. Due to long input sequences, the model does not adapt to strong dynamical changes. In this work, a simple parameter-based approach is developed to predict the intended behavior using the last seconds of the measured position variables. The idea is to globally identify the model parameters of the vessel, which remains constant for the situation, and additionally two parameters for local adaptation, which adapt at every updated input sequence. Typically parameters like rudder angle, wind velocities, and water current affect the behavior of vessels. The introduced approach works with a sliding window approach for which, after identification of the global system, local values are identified based on the last 80 measurements of the vessels. A trajectory prediction (assuming no additional rudder-based maneuvering) is realized for the prediction horizon of 180 s. To confirm the robustness of the new approach, real AIS/GPS-based measurements from a German research inland vessel for different scenarios and sailing conditions including 'loaded' and 'empty' sailing cases are used. Furthermore, additional results are shown for position data information of different sample rates.

**Keywords:** vessel model; trajectory prediction; AIS; vessel motion model; vessel position prediction

## 1. Introduction

Autonomous vessel sailing is becoming an attractive research topic in recent years. Due to a large number of vessel collisions and economical advantages, autonomous vessel sailing becomes attractive in companies and also for research. The accident statistics of the European Maritime Safety Agency state that human errors are the initial factor in 62 percent of cases with EU registered ships from 2011 to 2016 [1]. To avoid collisions in the case of autonomous sailing, trajectory prediction plays a significant role. To realize the trajectory prediction task, accurate prediction models are required.

Due to the nonlinear relations between hydrodynamics and complex rigid motion behavior of the ship, the modeling process is difficult. Models can be obtained using two principal approaches. Experimental modeling is economically costly, but accurate and reliable when tested well. The related methods are applicable only in areas/situations with

suitable and identical conditions. On the other hand, theoretical methods are based on equations that explicitly consider fluid mechanics, can be applied to a variety of situations, and are difficult to validate. Mostly computational fluid dynamics (CFD) techniques are used [2]. In [3], the authors use coefficient-based models that allow fast time simulations of arbitrary rudder maneuvers. The disadvantage of this approach is that it requires data from many system variables such as rudder angle, drift angle, yaw rate, dimensions of a ship, and non-dimensional hydrodynamic derivatives derived from an Abkowitz ship model using Taylor series [3]. It is known that simulation-based experimental modeling is a well-established approach for generating ship-specific models to be used for further examination. Therefore, it becomes obvious that the geometry of ships is required and complex simulations or experiments have to generated. The use of these kinds of models for prediction of the paths of encountering ships is not suitable due to missing knowledge like the unknown geometry of encountering ships (the dynamical behavior has to be predicted).

AIS (Automatic Identification System) is an automatic tracking system that identifies and locates vessels in the sea. This is achieved by data exchange with other ships, AIS base stations and satellites [4]. Thus, it can be said that the Automatic Identification System (AIS) is an international standard for ship-to-ship, ship-to-shore, and shore-to-ship exchange of information. This information includes vessel identity, position, speed, course, destination and other critical information, which ensures navigation safety and maritime security.

In the ship modeling literature [5], several identification methods for simplified ship parameter identification are suggested like least squares method, Bayesian approach, maximum likelihood method, extended Kalman filter [6], etc. These methods are suited to different hull forms and environment conditions. It is noted that the methods are sensitive to noise and initial conditions. To cope up with initial conditions, the authors in [7] use recursive least square (RLS) in combination with Support Vector Machine (SVM), assuming there are no disturbances acting on the system. In [5], the authors develop an approach to identify parameters under disturbances using SVM. The problem with this approach is that it cannot be applied to time-varying systems. The reason is that disturbances are assumed as constant, but in reality the ship dynamics are changing. Neural network approaches have been developed to identify the model parameters using ARX model [8]. Here, RLS provide initial identification results. Linear decreasing inertia weight particle swarm optimization is used to define optimal parameters by minimizing the global sum of square error (SSE) as an objective. These models do not require any initial estimates. On the other hand, in the case of black-box models, the parameters determined by neural networks are not necessarily correlated to physical properties of the ship. The effect of different environmental effects on trajectory prediction are not considered in this approach.

Many approaches have been used to identify the parameters. The main drawback of most of the approaches is that a large number of coefficients for ship dynamics, which are effected with regard to environmental disturbances and load conditions, have to be identified. In the case of encountering ships, the loading value, rudder angle, and yaw rate are typically unknown. As a result, the above mentioned approaches will provide models for specific cases or environment conditions.

In [9], the authors applied deep learning approaches to predict the positions of ships in the near environment with respect to their positions. The approach decomposes historical ship behaviors into local behavioral models. In [10], the authors used sequences of past AIS observations to produce future predictions using Recurrent Neural Networks. The drawback of the above mentioned approaches is that the local behavior model still considers the past data of 30 min with an update of local behavior every 15 min. It is assumed that this is too large for the consideration of short time dynamical changes of vessels.

To overcome the aforementioned restrictions in this contribution, an approach is developed to predict the trajectory in the near future based on position measurements only. Thus, the data are assumed as measurable from AIS systems or radar-based ship-based systems. The approach identifies the suitable parameters for the proposed model and takes the environmental disturbances implicitly into account. Additionally, local

model parameters are locally updated. Thus, in this contribution, the main focus is the development of a model-based trajectory prediction method. Here "model-based" refers to a local model generated by measurements. To ensure practical applicability, the robustness of the approach is addressed and validated.

The structure of the paper is as follows. In Section 2, the problem is described followed by the description of the approach taken in this work. Additionally, a measure for validation of the approach is given. In Section 3, the data used in this work and details of the ship used for experiments are described. To illustrate the robustness and applicability of the approach, different scenarios and situations are used, and related results are obtained and explained in detail. The AIS information is available in different sampling rates. The effects of different sampling rates of the information on trajectory prediction are evaluated in Section 3.3. Therefore, for the introduced scenarios and conditions, the assumed measurements are additionally artificially corrupted to evaluate the effect on the prediction capability of the new approach. Finally, a summary and the conclusions are presented.

## 2. System Identification Approach

System identification entails developing a mathematical model using the inputs and outputs of a system. Different approaches of system identification are known in the literature. These depend on whether the system is assumed as linear/non-linear or parametric/non-parametric. System identification involves five steps [11]: collecting the data, selecting the model structure either from prior knowledge or trial and error, choosing a criterion (cost function) to fit the model to the data, obtaining the parameters of the model by optimizing a suitable cost function, and finally, validating the model by testing it on new data (not used for "training"/identification). For the approach developed in this paper, it is assumed that the input position information for a certain part of the past trajectory is available as position measurement from the past trajectory. As output the upcoming position trajectory is predicted over a certain extent of the trajectory.

### 2.1. Problem Description

While determining the path of the vessel, collision avoidance is an important tasks for realizing safe operation of autonomous vessels. To ensure an acceptable level of safety of autonomous vessels, collision avoidance must also be efficient. The prediction of the upcoming behavior of objects in this context (typically other vessels/encountering vessels) is needed. Here, the trajectory prediction is also noted as behavioral intention prediction of other vessels. In this contribution, the environment can be rivers, channels, and ports with high traffic density. The smaller the area of movements (like the considered case of vessels sailing on rivers or channels), the more stringent the requirements. Model-based approaches independent of complex kinematic equations should be developed that are able to adapt to real environmental changes. To develop a ship model, the first step is to define the input variables. In this contribution, it is assumed that the measured variables are position variables generated by the AIS system (or alternatively generated by a ship-based radar system). For the application to serve in assistance systems as well in autonomous systems, the developed model should not be complex and must be applicable in real time.

### 2.2. Approach for Global and Local Model Identification

In this work, the task of defining the dynamical behavior of ships (ship model) is achieved in two steps. In the first step, the structure of the model has to be chosen. Different models of first-, second-, third-, and fourth-order systems are tested. The second-order system is selected based on better performance vs. the competing models. The model is defined for one coordinate of the position (longitude, latitude) using $y$ as output and $u$ as input with the equation

$$\ddot{y} + a_{22}\dot{y} + a_{21}y = b_1 u + k_1. \tag{1}$$

The model is transformed into a state space form as

$$\underbrace{\begin{bmatrix} \dot{y} \\ \ddot{y} \end{bmatrix}}_{\dot{x}} = \underbrace{\begin{bmatrix} 0 & 1 \\ -a_{21} & -a_{22} \end{bmatrix}}_{A} \underbrace{\begin{bmatrix} y \\ \dot{y} \end{bmatrix}}_{x} + \underbrace{\begin{bmatrix} 0 \\ b_1 \end{bmatrix}}_{B} \underbrace{u}_{u} + \underbrace{\begin{bmatrix} 0 \\ k_1 \end{bmatrix}}_{k}, \tag{2}$$

$$\tilde{y} = \underbrace{\begin{bmatrix} 1 & 0 \end{bmatrix}}_{C} \underbrace{\begin{bmatrix} y \\ \dot{y} \end{bmatrix}}_{x} + \underbrace{\begin{bmatrix} d_1 \end{bmatrix}}_{D} \underbrace{u}_{u}, \tag{3}$$

with

$A$—State matrix $2 \times 2$, $x$—State vector $2 \times 1$,
$B$—Input-to-state matrix $2 \times 1$, $u$—Input vector $1 \times 1$,
$C$—Output matrix $1 \times 2$, $y$—Output vector $1 \times 1$,
$D$—Transmission matrix $1 \times 1$, $a_{21}, a_{22}$—Stiffness and damping coefficients,
$k$—Vector of unknown inputs $2 \times 1$, and $b_1$—Coefficient of input affecting $\ddot{y}$.

Equation (2) assumes the input $u$ via matrix $B$ as well as an unknown input denoted as $k_1$ acting on the system. The resulting model is therefore determined by the global parameter matrices $A, B, C, D$ as well as the vector $k$ assumed as local adaptable.

The main idea of the proposed approach is to assume this mathematical model for both variables of the planar 2D-behavior of the vessel. The unknown matrices $A, B, C,$ and $D$ as well as $k$ have to be identified by a suitable online applicable procedure. Identification is the first step to obtain the parameters of $A, B, C, D$ in [12]. This identification process has to minimize the error between predicted output $\hat{y}$ as prediction for $\tilde{y}$ and measured position $y$. The problem is formulated as an optimization problem and solved using minimizing the Mean Square Error (MSE).

In this contribution, 2D positions (longitude, latitude) are considered using two models named as $M_{long}$ and $M_{lat}$, respectively. Here $M_{long}$ and $M_{lat}$ describe the models identified separately for longitude and latitude positions, respectively.

A sliding window approach is applied to identify the local parameter $k$. The window is defined as $\{y\}_{t_{k-h}}^{t_k} = f(\{x\}_{t_{k-h-l}}^{t_{k-h-1}})$, where $f(x)$ contains global parameters $A, B, C, D$ and local parameter $k$. Parameter $k$ is determined using the input sequence of positions $l$ and output sequence of positions $h$. The length of the output sequence $h$ is decided according to [13], where it is taken as 125 s. In this work, it is considered as 3 min. The local parameter is adapted using a window until time point $t_k$. The model performance depends consequently on $l$ and $h$. The predictions for the past $l$ input sequence are calculated using the already determined function $f(x)$ by $\{y\}_{t_{k+1}}^{t_{k+h}} = f(\{x\}_{t_{k-l+1}}^{t_k})$.

As illustrated, the model is composed of global (fixed) parameters $A, B, C, D$ and a local (locally adapted) parameter $k$. Thus, the local parameter $k$ is estimated online and is assumed time-varying. A sliding window approach is used to calculate the local parameter where a window of the specified length $l + h$ moves over the data at time $t_k$, with $t_{k-l+1}$ denoting the beginning of the past data interval and $t_{k+h}$ the prediction interval as shown in Figure 1.

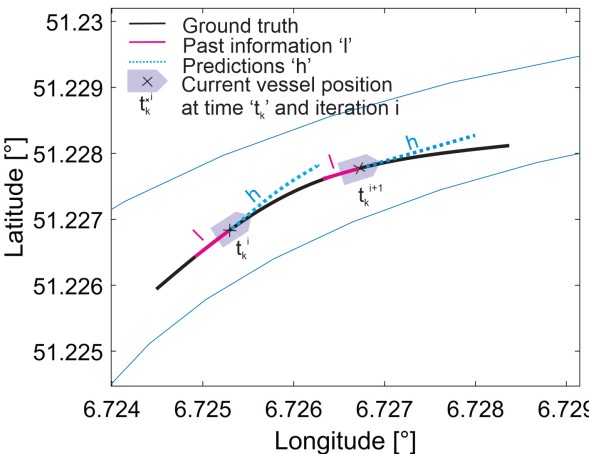

**Figure 1.** Example of local model updated iteratively using sliding window of *l* past sequence and *h* output sequence where $t_k^i$ and $t_k^{i+1}$ represent the time point of prediction where local parameter *k* is updated at iteration *i* and *i* + 1, respectively.

### 2.3. Measures for Validation

The average displacement error (ADE) metric is selected to evaluate the predictions. This refers to the mean square error (MSE) over prediction horizon *T* of the trajectory and the ground truth points as shown in Figure 2 and is calculated as

$$ADE(X_x, X_y) = \frac{1}{T} \sum_{k=1}^{T} \sqrt{\left(\hat{x}_{x,k} - x_{x,k}\right)^2}, \frac{1}{T} \sum_{k=1}^{T} \sqrt{\left(\hat{x}_{y,k} - x_{y,k}\right)^2}. \qquad (4)$$

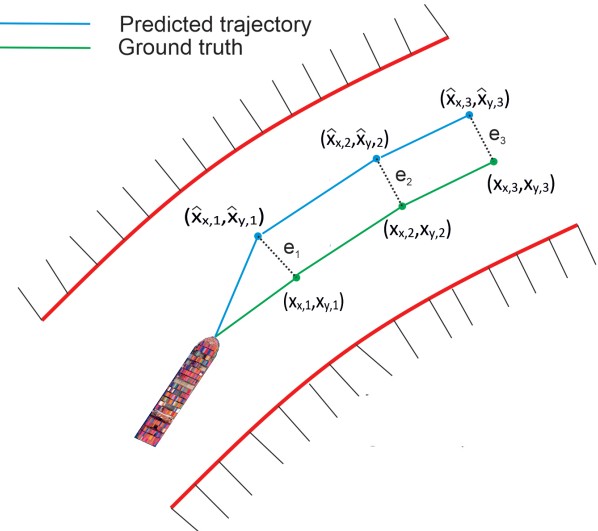

**Figure 2.** Side-to-side comparison of ground truth with prediction represented by dotted lines [14].

## 3. Experimental Evaluation and Approach Validation

### 3.1. Example: Used Material and Data

For the validation of the new approach, real measurements are used. Here, data from a German inland vessel as part of the 'PROMINENT' project [15] (promoting innovation in the inland waterways transport sector), funded by the European Union's Horizon 2020 Research and Innovation Program and provided by 'the Federal Waterways Engineering and Research Institute' (BAW), are applied.

### 3.1.1. Used Inland Vessel

The used data taken from a cargo vessel [16] is sailing under the flag of Germany. The length of the vessel is 135 m and the width is 14.2 m. The ship is equipped with sensors, GPS, flow velocities, echo, loading value, inclinometer, and depth sensors.

### 3.1.2. Validation Datasets (Prominent Datasat/AIS Dataset)

According to the International Maritime Organization (IMO), ships provide AIS, which provides information to other ships and related land-based stations. The data are transmitted through 27 message types. These messages include the navigational information, such as time, course over ground (COG), speed over ground (SOG), position, the IMO number of the ship, actual draft, departure, destination, flow velocity, etc. The dataset from the prominent ship is a time-series dataset of one year with a sampling rate of one second. The data contains the information of ship sailing in upstream/downstream, loaded/unloaded, and of different water levels. Realistic scenario dataset are using sampling rates of more than one second. The used dataset are generated on a research vessel so perfect AIS-GPS information are guaranteed. Additionally another AIS dataset is used. The difference between the two datasets is shown in Table 1. Here, only the usual AIS information are given and used. The real data are generated on board of the 'Niedersachsen 22' vessel in the context of the 'Autobin' project [17] using an average sampling rate of ten seconds.

**Table 1.** Difference between the prominent-dataset and AIS dataset.

| Parameters | Prominent Dataset | AIS Dataset |
| :---: | :---: | :---: |
| Position | ✓ | ✓ |
| Speed over Ground (SOG) | ✓ | ✓ |
| Course over Ground (COG) | ✓ | ✓ |
| Ship length | ✓ | ✓ |
| Flow velocity | ✓ | - |
| Depth | ✓ | - |
| Roll, Pitch | ✓ | - |
| Loading value | ✓ | - |
| Engine rpm | ✓ | - |
| Sample rate (in Hz) | 1 | 10 |

### 3.2. Effect of the Variation of Training to Prediction $(t/p)$ Ratio

The sliding window approach is explained in Section 2.2. Testing is needed to determine the input sequence $l$ for local parameter $k$ adaption for minimum prediction error. The training to prediction ratio $tn/p$ is selected to check the performance, where $p$ is the prediction horizon of 180 s. The input sequence $tn$ is selected as 40 s, 80 s, and 180 s. Training to prediction ratio is varied to analyze the effect on the model performance. Three different cases with the prediction ratios 2/9, 4/9, and 9/9 are selected. The results are shown in Figure 3. Here the ADE error between prediction and the ground trajectory is plotted. It can be concluded from Figure 3 that the different ratios do not affect model performance.

### 3.3. Scenario Description and Results

To confirm the robustness of the developed model and also the effect of hydrodynamical conditions, the model is tested on different scenarios. Considering the geometrical structure of the river, three scenarios are defined considering straight Figure 4a, curved Figure 4b and sharp curved, Figure 4c paths. The straight path from 703–710 km, curved path from 690–703 km, and sharp curved path from 735–745 km of the 'Rhine river' are considered. Based on the flow direction, the two cases "upstream and downstream" are considered. Two other cases defined by the load of the vessel as "loaded and unloaded" are considered. Thus, there are four cases, "upstream-loaded (case-1), downstream-loaded (case-2), upstream-unloaded (case-3), downstream-unloaded (case-4)", based on combina-

tions of loading, and flow cases. The model performance has to be tested in total for twelve cases (4 cases × 3 scenarios).

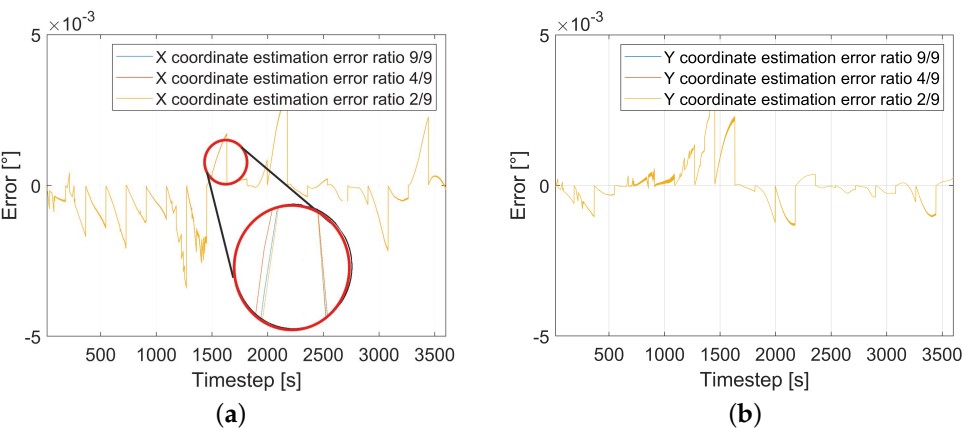

**Figure 3.** Model performance for varying training to prediction ratios: (**a**) $M_{long}$, (**b**) $M_{lat}$.

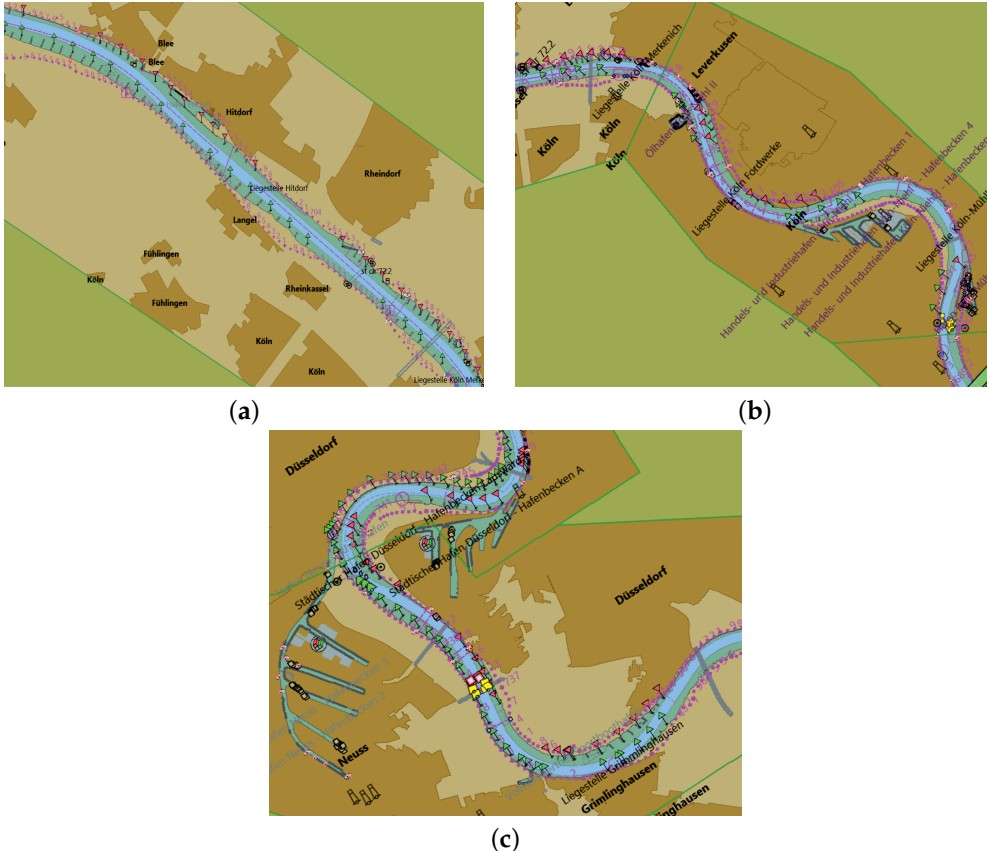

**Figure 4.** Three scenarios based on geometry of the Rhine river: (**a**) Straight path from 703–710 km taken from the Cologne region; (**b**) Curved path from 690–703 km from the Cologne region; and (**c**) Sharp curved path from 735–745 km representing the Neuss Hafen area.

### 3.3.1. Straight Path

In the straight path scenario, four different cases are considered. As mentioned in Section 2.3, ADE is calculated separately for both models $M_{long}$ and $M_{lat}$. The ADE results are explained (Figure 5) with two figures for $M_{long}$ in the first column and the other two figures for $M_{lat}$ in the second column. For every model, two figures with the first row representing the case-1 and 3 and the second row representing case-2 and 4 are used. The model performance is shown in Table 2. Here, the two rows represent different models,

the four columns represent the four cases. From Figure 5 and Table 2 it can be concluded that the ADE remains in the range of 100 m in most of the cases. The maximum error occurs in case 4 for a downstream-loaded situation.

**Table 2.** The maximum error of two models ($M_{long}$, $M_{lat}$) is shown for the four different cases. The green and red colored cases represent the best (green) and worst (red) results of each model in different cases.

| ADE | Case-1 | Case-2 | Case-3 | Case-4 |
|---|---|---|---|---|
| $M_{long}$ | 40 | 190 | 35 | 150 |
| $M_{lat}$ | 170 | 175 | 50 | 240 |

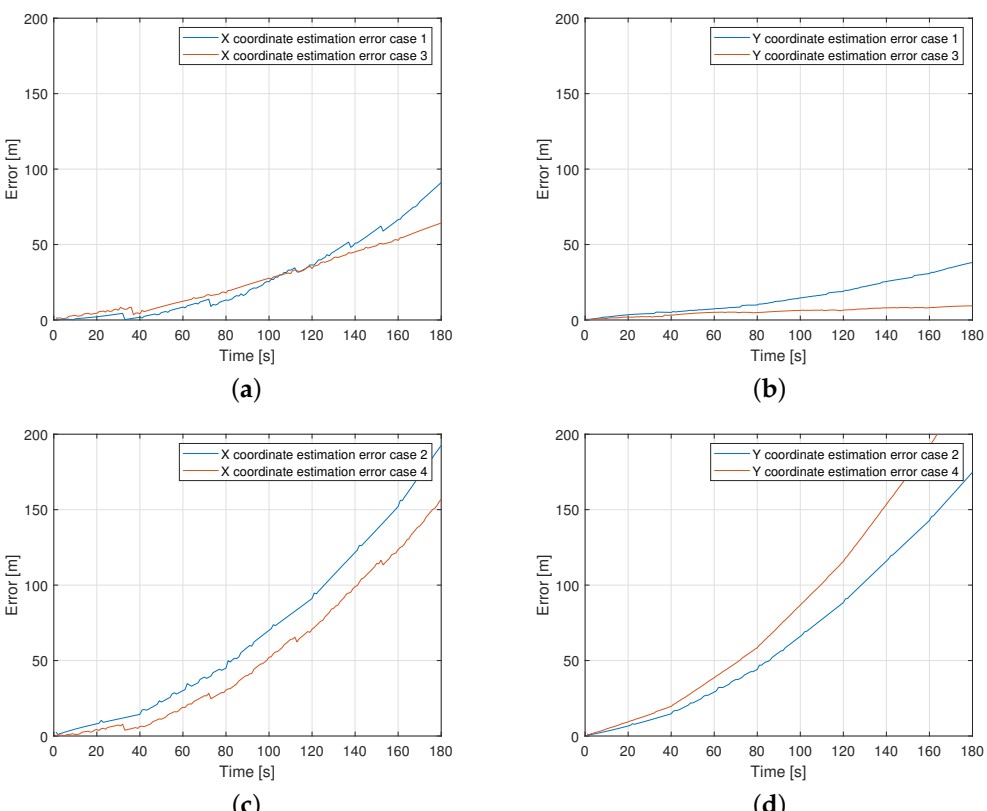

**Figure 5.** Model performance of straight path: (**a**,**c**) performance of $M_{long}$ in the upstream-loaded and upstream-unloaded scenarios; (**b**,**d**) performance of $M_{lat}$ in the downstream-loaded and downstream-unloaded scenarios.

### 3.3.2. Curved Path

In the curved path scenario, four different cases are considered. In this scenario, the complexity is increased in comparison to the straight path case. The model performance is shown in Table 3, where the two rows represent different models $M_{long}$ and $M_{lat}$, and the four columns represent the four cases. From Figure 6 and Table 3, it can be concluded that the ADE remains in the range of 60 m except $M_{long}$ for upstream, where it is more than 175 m.

**Table 3.** The maximum error of two models ($M_{long}$, $M_{lat}$) is shown for the four different cases. The green and red colored cases represent the best (green) and worst (red) results of each model in different cases.

| ADE | Case-1 | Case-2 | Case-3 | Case-4 |
|---|---|---|---|---|
| $M_{long}$ | 25 | 80 | 60 | 90 |
| $M_{lat}$ | 18 | 175 | 5 | 48 |

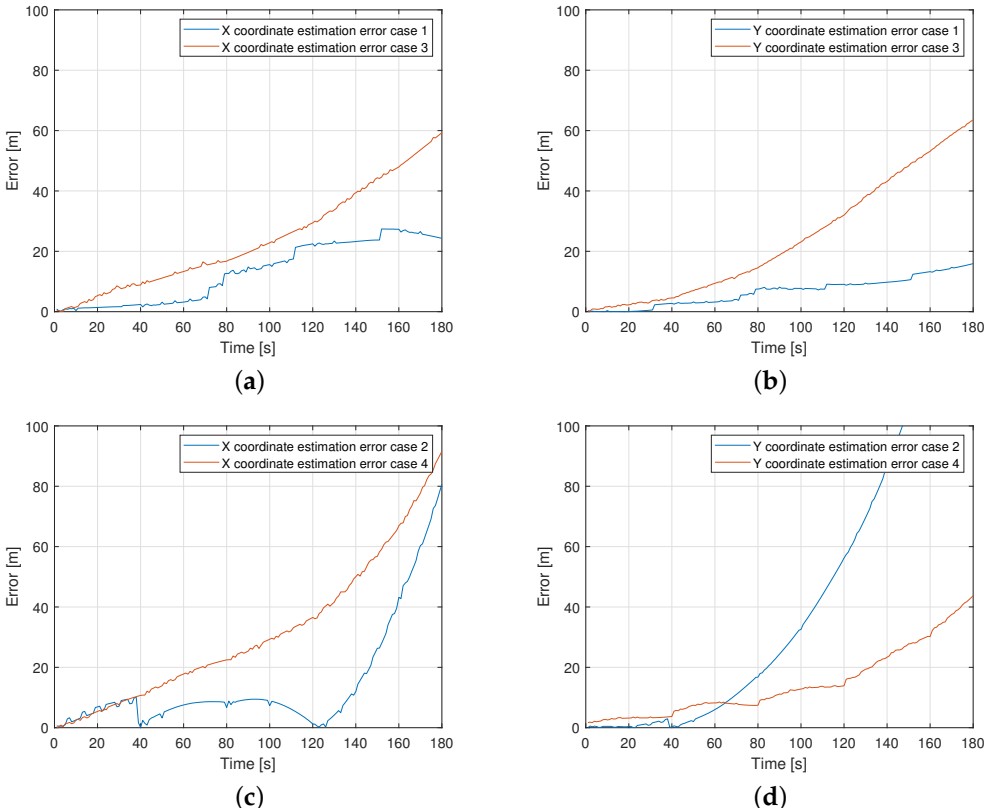

**Figure 6.** Model performance of curved path: (**a**,**c**) performance of $M_{long}$ in the upstream-loaded and upstream-unloaded scenarios; (**b**,**d**) performance of $M_{lat}$ in the downstream-loaded and downstream-unloaded scenarios.

### 3.3.3. Sharp Curved Path

In the sharp curved path scenario, the same four different cases are considered. This scenario has the highest complexity in comparison to straight and curved path cases. The model performance is shown in Table 4 where the two rows represent different models $M_{long}$ and $M_{lat}$ and the four columns represent the four cases. From Figure 7 and Table 4, it can be concluded that the ADE remains in the range of 100 m for all cases except for the upstream-unloaded with an error of 300 m.

**Table 4.** The maximum error of two models ($M_{long}$, $M_{lat}$) is shown for the four different cases. The green and red colored cases represent the best (green) and worst (red) results of each model in different cases.

| ADE | Case-1 | Case-2 | Case-3 | Case-4 |
|---|---|---|---|---|
| $M_{long}$ | 190 | 170 | 125 | 50 |
| $M_{lat}$ | 50 | 170 | 300 | 45 |

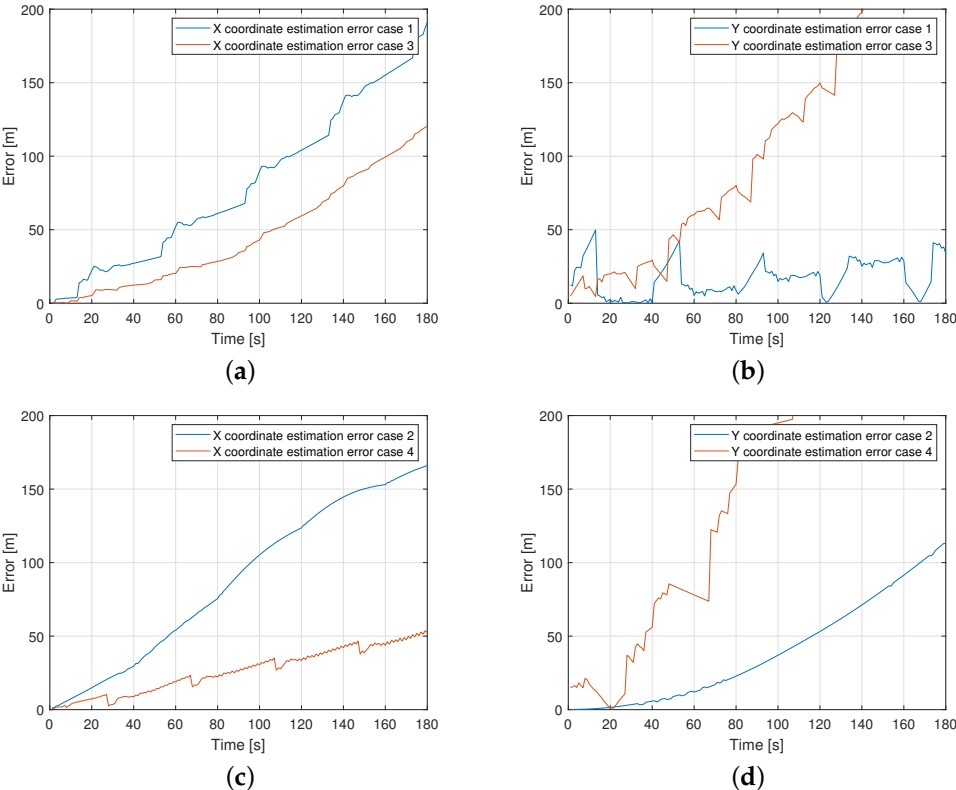

**Figure 7.** Model performance of sharp curved path: (**a**,**c**) performance of $M_{long}$ in the upstream-loaded and upstream-unloaded scenarios; (**b**,**d**) performance of $M_{lat}$ in the downstream-loaded and downstream-unloaded scenarios.

### 3.4. Effects and Discussion of Different Sampling Rates of AIS-Data

As illustrated the 'PROMINENT' data set is precise in comparison to real ships real AIS data. Real AIS-data are sent in intervals of 3 or more seconds, depending on the speed of the vessel. As a conclusion, it can be assumed that the loss of data precision with reduced frequency/less number of position information transmitted in comparison to the 'Prominent' data used affects the prediction accuracy. So, real AIS-data can be simulated from the Prominent dataset with sampling rate information available every 2 s, 5 s, 10 s to consider different AIS receiver characteristics. The data are generated by selecting the data points every 2, 5, 10 from the higher sampled dataset respectively. Furthermore, AIS-data from Niedersachsen 22 (N22) vessel [17] are additionally used. The AIS data are available with an average sample rate of 10 s. The data are collected from the Dortmund-Ems channel; therefore, only two scenarios, the straight path and sharp curved path, are considered. It is tested how the information with different sampling rates affects the model performance. The performance criteria ADE is used.

In this case, only combinations of loaded vessels are considered for downstream/upstream situations on a channel in combination with the two scenarios, straight and sharp curved path, being considered. The model performance is tested as shown in Figures 8 and 9 and Table 5. It can be seen that in both scenarios, the upstream situation has less prediction errors than in the downstream situation. The reason behind this results from the fact that the vessel in inertial coordinates is slower in upstream situation. It can be stated that the error increases with a high sampling rate of information, but not so significantly.

The results of the model using the N22 data are shown in Figure 10. It can also be stated that the error is almost negligible in a straight curve, but the sharp curved path scenario has a maximum error of 0.004. It can also be detected that data from the Prominent dataset are very precise, so the error is less compared to those from the more realistic case (N22).

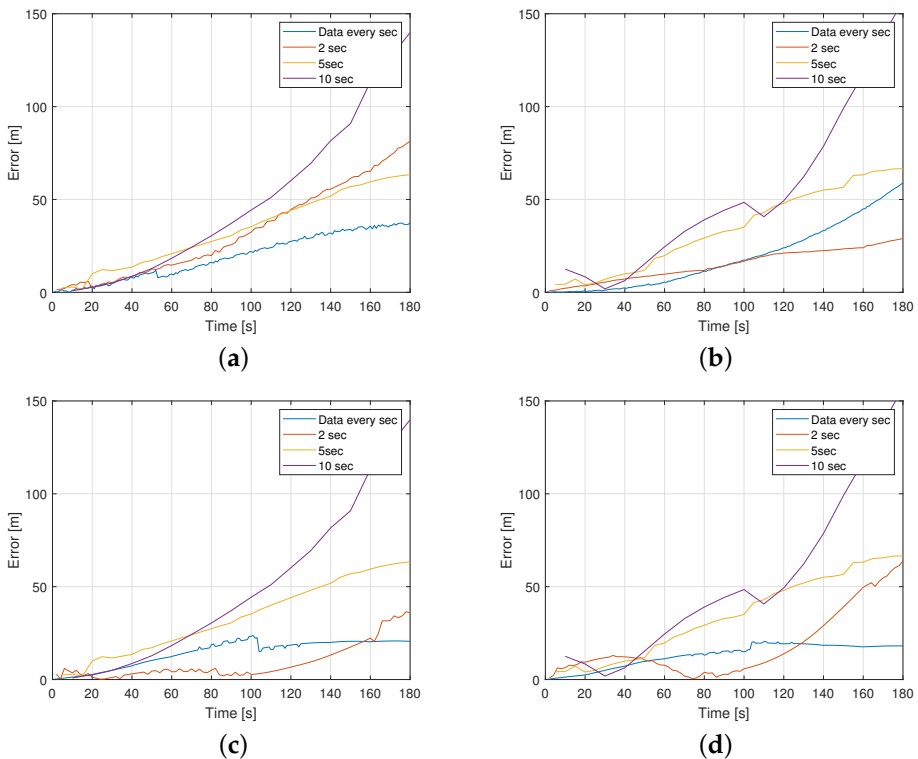

**Figure 8.** Model performance for different information update sample rate [1, 2, 5, 10] s; here: straight path: (**a**,**c**) performance of $M_{long}$ and (**b**,**d**) performance of $M_{lat}$ y-coordinate in the upstream- and downstream-loaded scenarios, respectively.

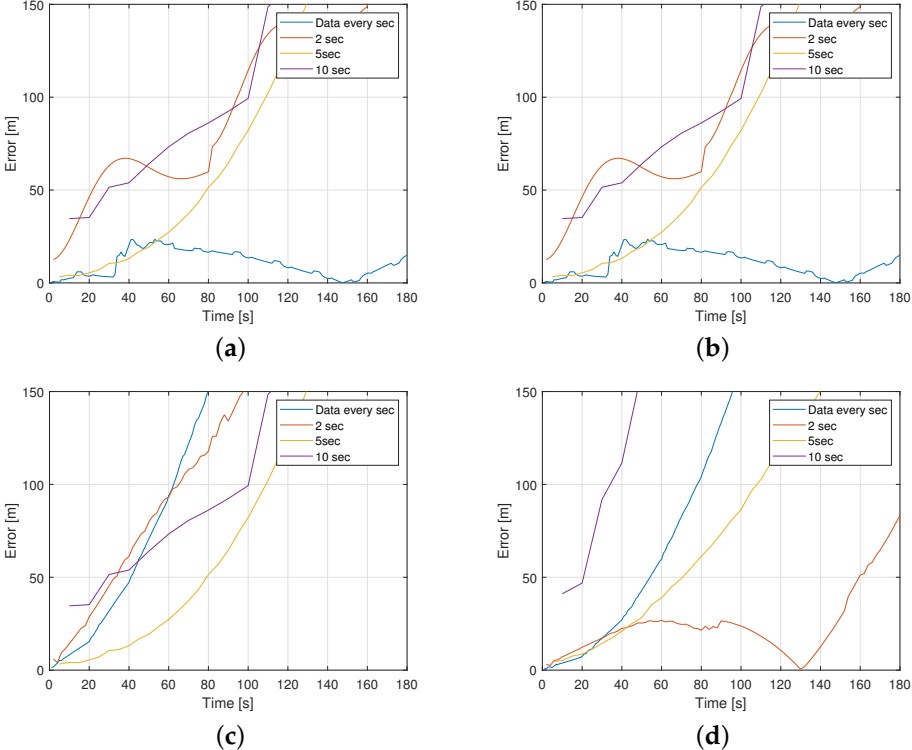

**Figure 9.** Model performance for different information update sample rates [1, 2, 5, 10] s; here: sharp curved path: (**a**,**c**) performance of $M_{long}$ and (**b**,**d**) performance of $M_{lat}$ y-coordinate in the upstream- and downstream-loaded scenarios, respectively.

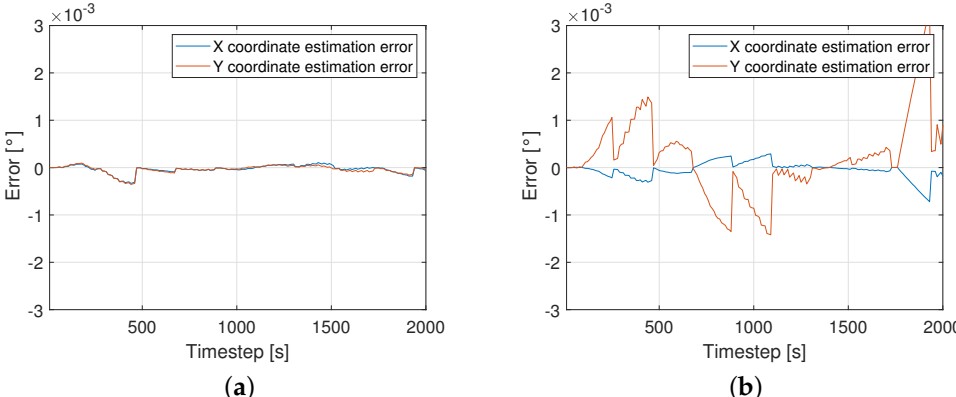

**Figure 10.** Model performances of $M_{long}$ and $M_{lat}$ in case of Niedersachsen 22. (**a**,**b**) performance of straight and sharp curved path, respectively.

**Table 5.** Mean error of two models ($M_{long}$, $M_{lat}$) is shown for the four different situations and different sampling rates of information. The green and red colored cases represent the best (green) and worst (red) results of each model in different cases.

| Scenarios | ADE | Every Second | 2 s | 5 s | 10 s |
|---|---|---|---|---|---|
| Straight path | Upstream $M_{long}$ | 50 | 80 | 130 | 30 |
| | Downstream $M_{lat}$ | 50 | 80 | 130 | 30 |
| | Upstream $M_{long}$ | 18 | 210 | 220 | 48 |
| | Downstream $M_{lat}$ | 40 | 80 | 135 | 18 |
| Sharp curved path | Upstream $M_{long}$ | 25 | 40 | 60 | 140 |
| | Downstream $M_{lat}$ | 60 | 25 | 20 | 15 |
| | Upstream $M_{long}$ | 20 | 50 | 70 | 140 |
| | Downstream $M_{lat}$ | 16 | 68 | 70 | 160 |

## 4. Summary and Conclusions

In this contribution the trajectory prediction of vessels is provided. In contrast to the usual approaches, here a simple parameter-based approach is developed and validated using measured position information from different vessels (Prominent dataset, Niedersachsen 22). A sliding window approach is applied to continuously identify the model. The model is used to predict the upcoming trajectory of encountering ships and therefore their assumed intentions. The approach presented a model for a local time scale (prediction horizon of 180 s) using global parameters identified through a middle time scale. The key idea is to adapt a single parameter on a very local/narrow time scale to represent local effects affecting the dynamical behavior of vessels. For validation different scenarios varying in their complexity are used. The results show that the newly introduced approach solves the given task. As an advantage, an easy to identify model that allows an initial prediction using easy to have position measurements is established. The results indicate in detail that the model performance depends on the different situations as well as the geometry of the river. Estimates for upstream scenarios are more feasible than those with downstream vessels because the vessel moves more slowly. Furthermore, the results show that model predictions for measurements available every 1 or 2 s show very good results. For data availability that is updated only 10 s, on the other hand, the model has to be continuously re-trained, so the adaptation of the local parameter has to be adjusted earlier.

The presented approach shows the possibilities and limitations of using an easy-to-realize data-driven prediction algorithm, which in combination with the behavior estimation of encountering vessels will be essential for autonomous ships. This model will later be combined with other data-driven approaches generated from clustered historical data representing long-term behaviors of inland waterway vessels.

**Author Contributions:** Conceptualization, N.S.T. and D.S.; methodology, N.S.T. and D.S.; software, J.H.; investigation, N.S.T., J.H. and D.S.; resources, D.S.; writing—original draft preparation, N.S.T.; writing—review and editing, N.S.T. and D.S.; visualization, N.S.T. and D.S.; supervision, D.S.; project administration, D.S. All authors have read and agreed to the published version of the manuscript.

**Funding:** We acknowledge support from the European Regional Development Fund (ERDF), grant-no. EFRE-0801714.

**Data Availability Statement:** Data are generated within the EU Horizon 2020 project PROMINENT [18] and are provided with restricted rights by BAW. Details about the project can be found in [16].

**Acknowledgments:** We acknowledge support by the Open Access Publication Fund of the University of Duisburg-Essen.

**Conflicts of Interest:** The authors declare no conflict of interest.

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
