# Peer review of "Fast and Precise Generic Model for Position-Based Trajectory Prediction of Inland Waterway Vessels"

_2673-4052, doi:10.3390/automation3040032_

Round 1
Reviewer 1 Report
1)“Thus, there are four cases ‘upstream-loaded (case-1),downstream-loaded (case-2), upstream-unloaded (case-3), downstream-unloaded(case-4)’ based on combinations of loading, and flow case.”
There are four cases in this work. If the AIS data set were trained separately? If not, what is the benefit of these four cases?
2) In recent years, many papers have been published on trajectory prediction based on deep learning algorithms. Considering that most references are outdated, I propose to include more new theses about deep learning algorithms in the introduction and explain the superiority or novelty of this work.
3)Please include a detailed description of the training methods in this paper.
4)Equations 2.2-2.4 don't give a specific meaning for each variable, such as u1, d1, bs.
5) What is interesting about this work is the training of global parameters and the real-time calculation of local parameters, but the method is not discussed in detail in this paper.
6) Please describe the reason for choosing the prediction ratios 2/9, 4/9 and 9/9.
Reviewer 2 Report
This is an interesting study. It will play an important role to stop collisions in the case of autonomous sailing. The author's main work is on the prediction of trajectory. Several articles are available in the literature for trajectory prediction. Therefore, how we can justify the novelty of the proposed research work? My suggestion for revisions are as follows;
1. Research article has several mistakes from the language point of view, such as sentence structure, spelling mistakes, etc. The paper must be proofread by a native speaker.
2. References are not enough. It is suggested to cite more relevant and recent literature articles.
3. A comparison of the proposed work with published literature work is very important to verify the effectiveness of the proposed work. Published literature work must be from computational fluid dynamics and coefficient-based models-related work.
4. P2 Line 83 and P2 Line 85, what do chapters 2 and 3 refer to? Do they refer to some references?
5. The structure of the paper is not organized well. The “design methodology” part and “result and discussion” part has been merged into each other. A well-organized paper must consist of a design methodology and result discussion part with a clear boundary.
6. P3, Line 122, Equation 2.1, the author has taken a second-order differential equation as a structure of the model. The author did not explain why he chose the second order while a model with higher order equation can have better results.
7. It is conventional to use capital letters to represent a matrix in state space while the author has used small letters to represent a matrix in state space. Refer to equations 2.2 and 2.3.
